# Biomimetic Prosthetic Hand Enabled by Liquid Crystal Elastomer Tendons

**DOI:** 10.3390/mi12070736

**Published:** 2021-06-23

**Authors:** Haiqing Lu, Zhanan Zou, Xingli Wu, Chuanqian Shi, Yimeng Liu, Jianliang Xiao

**Affiliations:** 1College of Mechanical Electrical and Vehicle Engineering, Weifang University, Weifang 261061, China; luhaiqing668@163.com; 2Department of Mechanical Engineering, University of Colorado, Boulder, CO 80309, USA; zhanan.zou@colorado.edu (Z.Z.); wuxingli40682@163.com (X.W.); shichuanqian@tongji.edu.cn (C.S.); yili2293@colorado.edu (Y.L.); 3College of Mechanical Engineering, Shenyang University of Technology, Shenyang 110870, China; 4School of Aerospace Engineering and Applied Mechanics, Tongji University, Shanghai 200092, China

**Keywords:** liquid crystal elastomer, electric current, strain, stress

## Abstract

As one of the most important prosthetic implants for amputees, current commercially available prosthetic hands are still too bulky, heavy, expensive, complex and inefficient. Here, we present a study that utilizes the artificial tendon to drive the motion of fingers in a biomimetic prosthetic hand. The artificial tendon is realized by combining liquid crystal elastomer (LCE) and liquid metal (LM) heating element. A joule heating-induced temperature increase in the LCE tendon leads to linear contraction, which drives the fingers of the biomimetic prosthetic hand to bend in a way similar to the human hand. The responses of the LCE tendon to joule heating, including temperature increase, contraction strain and contraction stress, are characterized. The strategies of achieving a constant contraction stress in an LCE tendon and accelerating the cooling for faster actuation are also explored. This biomimetic prosthetic hand is demonstrated to be able to perform complex tasks including making different hand gestures, holding objects of different sizes and shapes, and carrying weights. The results can find applications in not only prosthetics, but also robots and soft machines.

## 1. Introduction

Prosthetic hands (Figure 1a [1]), as one of the most important implants serving amputees, have attracted a lot of research effort in the last few decades. However, the ideal characteristics required for prosthetic hands, including high performance, lightweight, bionic look, and affordability [2,3,4,5], are still lacking. Various improvements have been implemented, leading to Vincent hand [6], ILimb [7], Bebionic hand [8], and Michelange hand [9]. All these commercial prosthetic hands are still based on mechanical designs that utilize stiff joints and low degrees of freedom. This limitation leads to the bulky, heavy, complex and expensive products. To overcome these challenges, new mechanical designs have been investigated. For example, a tendon driven design has been introduced and was able to mimic the natural movements of human hands [10,11,12,13,14,15,16,17,18,19]. However, such design still depends on DC motors, as the actuation mechanism, and thus weight, performance, and affordability issues cannot be resolved.

The key resolution to these challenges could lie in the mechanisms of actuation; therefore, utilizing smart materials as the actuation mechanisms has been a focus of research. For example, a prosthetic arm equipped with a thermal pneumatic artificial muscle was successfully demonstrated [20,21,22,23,24,25,26]. Other pneumatics-powered health care and aid devices have been developed as well [27]. Shape memory alloys were also shown to be promising actuators for prosthetic hands and other parts [28,29,30,31,32,33,34]. More recently, dielectric elastomers were integrated into prosthetic systems, and have shown good performances [35,36,37,38,39,40,41]. Other examples include a nylon-based thermal actuator [42], a haptic actuator [43], and pneumatic actuators [44,45,46,47,48,49]. Among all these smart materials, liquid crystal elastomers (LCEs) can provide actuation in ways most similar to natural muscles, including large deformation, linear actuation, and fiber-like form factor, and, therefore, could be a good candidate for the next-generation actuation mechanism for prosthetics [50,51,52,53,54,55,56,57,58,59,60,61,62,63,64,65,66,67,68]. When synthesized, the LCEs are in the polydomain state. After being programmed using the stretching method, they transition to the monodomain state. During actuation, heating up the LCEs causes the transition from the monodomain state back to the polydomain state, leading to contraction deformation [69,70,71,72,73,74,75,76,77,78,79,80]. Then, cooling down changes the LCEs back to their programmed monodomain state, leading to extension deformation. Due to their excellent mechanical characteristics, previous studies have shown interesting applications of LCEs, including actuation [81], cell culture [82] and tissue repair [83].

In this paper, we present a material system based on liquid metal (LM) as the heater material and LCE as the soft actuator, and we demonstrate a biomimetic prosthetic hand actuated by LCE tendons. The LCEs were prepared using procedures described in previous studies [68,84,85,86]. Each tendon was constructed by sandwiching a liquid metal heating element (25% gallium and 75% indium) between a very high bond (VHB) (VHB, 3M) tape and an LCE film. Different from conventional metal heating elements, the liquid metal heating element can be deformed extremely together with the matrix, without increasing the stiffness or introducing resistance to the artificial muscle deformation, and thus is far more superior. Characterizations show that the LCE tendon can generate as much as 43.6% contraction strain and 536 kPa contraction stress during actuation, enabling a wide range of actuation modes in the prosthetic hand. We demonstrate that a biomimetic prosthetic hand driven by such LCE tendons can accomplish complex and delicate tasks, including showing complex hand gestures, holding objects of different sizes and shapes, and lifting weights.

## 2. Results

Figure 1b shows the anatomy of a finger of a human’s hand [87]. Its flexion motion is driven by the flexor tendon, protected by the sheath and held by the pulleys. Inspired by the human’s finger, a biomimetic prosthetic hand is presented in Figure 1c. The flexion of the fingers is driven by liquid crystal elastomer (LCE) tendons, held by silicone pulleys. An exploded view of the LCE tendon design is demonstrated in Figure 1d. It consists of a very high bond (VHB, 3M) tape support and bonding layer (10 mm wide and 25 µm thick) at the bottom, a liquid metal (LM) (25% gallium and 75% indium) heating element (1 mm wide and 0.5 mm thick) in the middle, and an LCE layer (1 mm thick) on the top. Copper wires are used to connect the LM heating element with an external power source. The fabrication of an LCE tendon started with patterning an LM heating element on top of the sticky side of a VHB layer using screen printing method. Two copper wires were connected at the end of the LM heating element. The LCE was prepared using a two-stage thiol-acrylate Michael addition-photopolymerization (TAMAP) reaction. 4-bis-[4-(3-acryloyloxypropypropyloxy) benzoyloxy]-2-methylbenzene (RM257, 4 g), pentaerythritol tetrakis(3-mercaptopropionate) (PETMP, 0.217 g), 2,2-(ethylenedioxy) diethanethiol (EDDET, 0.9157 g), (2-hydroxyethoxy)-2-methylpropiophenone (HHMP, 0.0257 g) and dipropylamine (DPA, 0.568 g) were dissolved in toluene solution, and casted on top of the VHB bonding layer and LM heating element. After curing at 80 °C for 12 h, the sandwiched structure was pre-stretched from the original length L_1_ to a programming length L_2_, as illustrated in Figure 1e. UV light was then used for fixing the programed state, which completed the fabrication of an LCE tendon. When an electric current is applied through the copper wires, the LM heating element heats up the LCE, leading to shrinking of the LCE tendon to its original length L_1_. When cooled down, the LCE tendon returns to the programmed length L_2_. In the following, the LCE tendons were prepared with an original length of L_1_ = 40 mm and were stretched by 200% to L_2_ = 120 mm during programming. In the design of the LCE tendon, because the thickness of the VHB layer (25 µm) is much smaller than that of the LCE (1 mm), and the LM heating element is in liquid state, their mechanical resistance to mechanical actuation is negligible, and therefore, the LCE tendon can actuate in a linear fashion, similar to flexor tendons in human fingers.
Figure 1Design and fabrication of LCE tendon and biomimetic prosthetic hand. Comparison of the design and structure between (**a**) a robot hand [1], (**b**) a human hand [87], and (**c**) the biomimetic prosthetic hand. (**d**) The design and multilayer structure of the LCE tendon. (**e**) Programming of the LCE tendon during fabrication process.
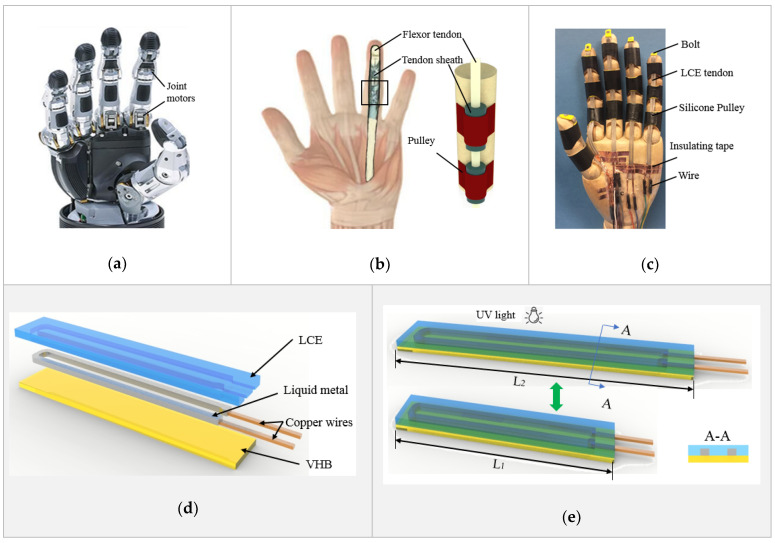



The LCE tendon was characterized using an Instron mechanical testing system (MTS), and a thermal camera (FLIR-T630) was placed nearby to record the temperature change and actuation strain, as shown in Figure 2a. A current generator was used to supply constant and stable electric current to the LM heating element. Joule heating led to phase transformation and contraction of the LCE tendon. In the first characterization, in order to measure the contraction strain in the LCE tendon, one end of the LCE tendon was fixed on the clamp while the other end was kept free. When enough current passed through the LM heating element, temperature increase caused the LCE tendon to shrink freely from the programming length L_1_ to the original length L_2_, as shown in Figure 2b and Appendix A. The thermal camera recorded both the actuation strain and temperature distribution on the surface of the LCE tendon. In the experiment, three different current values were supplied: 1.5 A, 2 A, and 2.5 A, and at least three tests were carried out for each current. The temperature in LCE tendon versus time for three currents is presented in Figure 2c, which shows the maximum temperatures that can be reached for each current, and the time required to reach the maximum temperature. For 1.5 A, 2 A and 2.5 A currents, the maximum temperatures that can be reached are 98 °C, 141 °C, and 160 °C, and the times required to reach these temperatures are 65 s, 50 s, and 30 s, respectively. The current was then cut off after the LCEs reached the maximum temperatures. The cooling of LCE tendons after heating was similar for the three different currents. Figure 2d exhibits the maximum contraction strain in each sample versus the supplied current. The average contraction strain increases from 26.5% at 1.5 A, to 40% at 2.0 A and 43.6% at 2.5 A. This clearly demonstrates the effect of increasing current on increasing the contraction strain of the LCE tendon in a shorter time.

A second characterization was carried out with both ends of the LCE tendon fixed on the clamps of the MTS. When electric current was supplied to the LM heating element, temperature increase-induced LCE contraction was prevented by the MTS clamps, and the contraction force/stress was recorded by the MTS. The contraction stress versus heating time for the three different currents is shown in Figure 2e. When the current increased from 1.5 A to 2 A, the maximum stress increased from 352 kPa to 546 kPa, and the time required to reach the maximum stress decreased from 109 s to 42 s. Further increasing the current to 2.5 A meant that the time required to reach maximum stress decreased to 21 s, but the maximum stress that could be reached decreased to 483 kPa. This is because a vast amount of heat was generated at 2.5 A, leading to debonding of the VHB layer with LCE, as well as localized polymer burning of the LCE artificial muscle.

There are situations that need the fingers of a prosthetic hand to stay at certain deformed configurations for a period of time, such as showing gestures, holding an object and lifting weights. These situations require the LCE tendon to be able to stay at a contracted state or to apply a constant force for a period of time. In order to achieve this, we explored a strategy to alternately control on and off of the electric power supplied to an LCE tendon. From Figure 2c, it is clear that the higher the electric current is, an LCE tendon heats up faster and thus reaches the maximal contraction strain and stress faster. Our experimental results indicate that an LCE tendon can easily break if the temperature reaches 120 °C for a long period of time. Therefore, the temperature of LCE tendons was kept at 110 °C in these tests. At this temperature, the performance of an LCE tendon can meet the requirements of most operating conditions. Additionally, in order to accelerate the heating of LCE tendons in order to achieve fast actuation, an electric voltage of 6 V and current of 4.5 A were used. As shown in Figure 3a, after 2 s of heating, the temperature of the LCE tendon reached ~120 °C. Then, turning off and on the electric power for 1.6 and 0.15 s alternately and repeatedly (see current vs. time control strategies in Figure 3c), the LCE tendon temperature stabilized at ~110 °C (2-1.6-0.15 curve in Figure 3a). While keeping the initial heating time at 2 s and the alternate power on time at 0.15 s, the alternate power off time of 1.5 s and 1.4 s were also used (2-1.5-0.15 and 2-1.4-0.15 curves in Figure 3a). It is shown that the temperatures for these two strategies decreased after initial 2 s heating, but then gradually increased again. The contraction stresses of the LCE tendons using these alternate power on and off strategies are depicted in Figure 3b. It clearly shows that the contraction stress of the 2-1.6-0.15 strategy stabilized at ~390 kPa, while the contraction stresses of the 2-1.5-0.15 and 2-1.4-0.15 strategies gradually increased with time. These results suggest that the 2-1.6-0.15 strategy can provide stable temperature and contraction stress in the LCE tendon.

While heating in an LCE tendon can be accelerated by increasing the electric power, cooling is a bottleneck for LCE tendons to achieve fast actuation. In order to accelerate the cooling in LCE tendons, we investigated the temperature decrease in LCE tendons using three different cooling methods, i.e., natural cooling, fan cooling and compressed air cooling. In these tests, the LCE tendons were heated up using an electric voltage of 6 V and current of 4.5 A. When the temperature of an LCE tendon reached 110 °C, the electric power was cut off. The LCE tendon was then cooled down by natural cooling, fan cooling or compressed air cooling. The LCE tendon temperature versus time using these three different cooling methods are shown in Figure 4a. Under natural cooling, it takes 102 s for the LCE tendon temperature to drop from 110 °C to 27.3 °C. Under fan cooling with a wind speed of 2.9 MPH (measured using an anemometer, Ambient Weather WM-2), the time needed for the LCE tendon temperature to drop from 110 °C to 27 °C decreased to 52.5 s. Compressed air cooling (Falcon Dust, Model: DPSJB) could further significantly reduce the time needed for the same temperature drop in an LCE tendon to 5 s. For all these cases, the LCE tendons restored to their initial length (130 mm) when their temperatures dropped to room temperature (27 °C), as shown in Figure 4b. These results demonstrate cooling strategies for either accelerating the cooling of LCE tendons for fast actuation or controlling the actuation in a programmable fashion.

One of the basic functions of a prosthetic hand is to make different hand gestures, which requires the independent control of each finger. Here, we embedded one LCE tendon into each finger, which was confined by three silicone “pulleys”, as shown in Figure 5a. For simplicity, rubber bands were used as “extensor tendons” for the recovery of the fingers to their original configurations, as presented in Figure 5a, right frame. When the current was applied to all five tendons, all five fingers bent to make a “fist” (Figure 5a, right). Other hand gestures including “thumb up” (Figure 5b), “OK” (Figure 5c), “I love you” (Figure 5d), “weak” (Figure 5e) and “victory” (Figure 5f) can also be realized by controlling the LCE tendons in the prosthetic hand. For all of these cases, when the electric power was cut off, the LCE tendons cooled down and extended, and the prosthetic hand returned to the initial state due to the restoring force provided by the rubber bands.

In addition to the hand gestures, the prosthetic hand can perform more complex tasks, such as picking up and grabbing objects, as shown in Figure 6. Firstly, we demonstrated that the prosthetic hand was able to hold different shapes of objects. For example, the prosthetic hand can bend all five fingers to conformally hold a soft paper tube of diameter 30 mm (Figure 6a), use the tips of the thumb, the middle and little fingers to grab a plastic beaker (Figure 6b), and coordinate the thumb and index fingers to pick up a maker pen of diameter 20 mm (Figure 6c, Appendix A). Furthermore, we also tested the load carrying capability of the prosthetic hand. As shown in Figure 6d, the prosthetic hand can gently hold a light-weight plastic tube with a diameter of 35 mm and weight of 2.12 g. Figure 6e demonstrates the prosthetic hand can carry a water bottle with a weight of 200 g by coordinately bending all five fingers (also see Appendix A). Figure 6f presents a demonstration of lifting a bag with a weight of 252 g with four fingers (also see Appendix A).

## 3. Conclusions

In this work, a novel artificial tendon was realized by combining an LCE and an LM heating element. The responses of the LCE tendon to different electric currents, including the temperature increase, contraction strain and contraction stress, were characterized. The strategies of achieving stable temperature and contraction stress in LCE tendons, and accelerated cooling were also explored. Then, the LCE tendons were integrated with a wood hand to achieve a biomimetic prosthetic hand that can perform various tasks, including making complex hand gestures, holding objects of different sizes and shapes, and carrying weights. Such capabilities of LCE tendons could find applications in not only prosthetics, but also robotics and soft machines. However, there are still drawbacks of the current LCE tendons that need further investigation before they can be practically applied, including slow response, high temperature actuation, and low energy efficiency.

## Figures and Tables

**Figure 2 micromachines-12-00736-f002:**
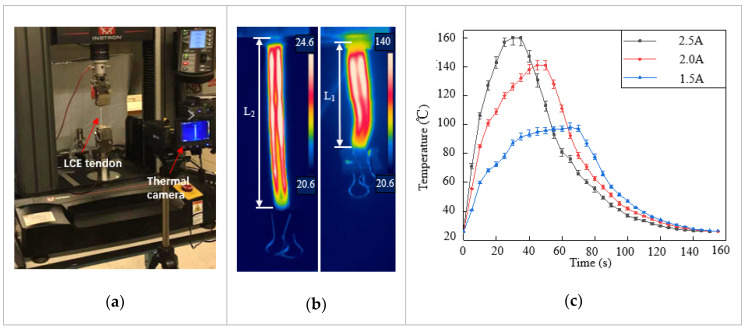
Characterization of LCE tendon. (**a**) Experimental setup for LCE tendon characterization. (**b**) Temperature distribution and length change of an LCE tendon at the initial state (left, room temperature) and full contraction state (right, heated). (**c**) The LCE tendon surface temperature versus time during joule heating due to different electric currents. (**d**) The maximum contraction strain can be reached in LCE tendons at different electric currents. (**e**) The contraction stress in LCE tendon versus heating time.

**Figure 3 micromachines-12-00736-f003:**
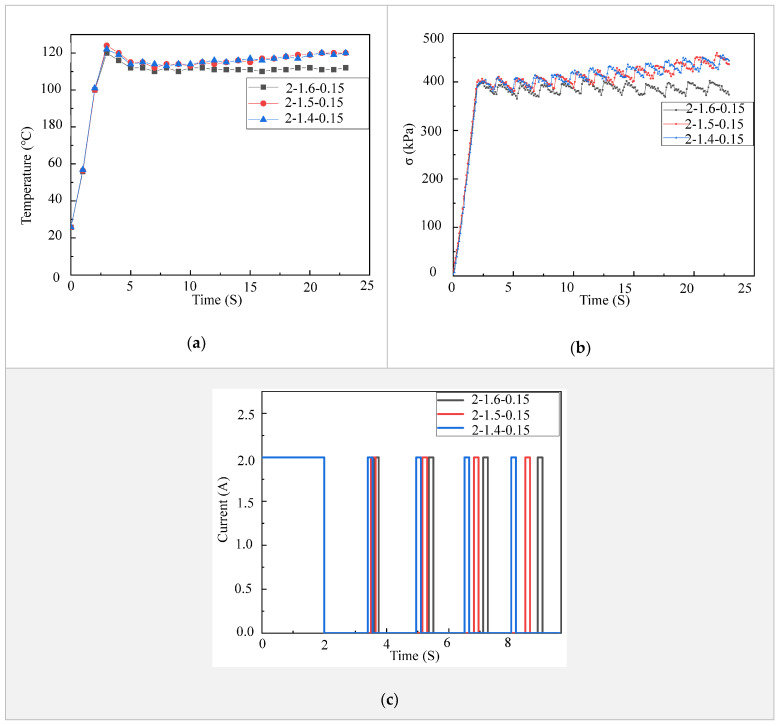
Strategies of achieving stable LCE tendon temperature and contraction. (**a**) LCE tendon surface temperature versus time using different control strategies. (**b**) LCE tendon contraction stress versus time using different control strategies. (**c**) Current versus time control strategies.

**Figure 4 micromachines-12-00736-f004:**
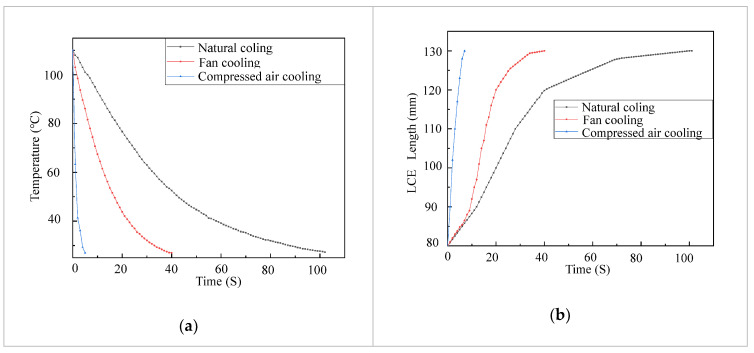
LCE tendon surface temperature drop (**a**) and extension (**b**) versus time using different cooling methods.

**Figure 5 micromachines-12-00736-f005:**
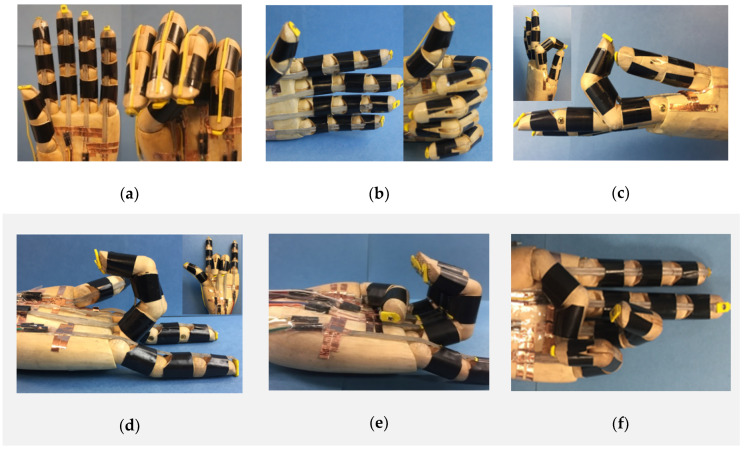
Hand gestures made by the biomimetic prosthetic hand. The biomimetic hand can coordinately control the fingers through the LCE tendons to realize various hand gestures, including (**a**) “fist”, (**b**) “thumb up”, (**c**) “OK”, (**d**) “I love you”, (**e**) “weak” and (**f**) “victory”.

**Figure 6 micromachines-12-00736-f006:**
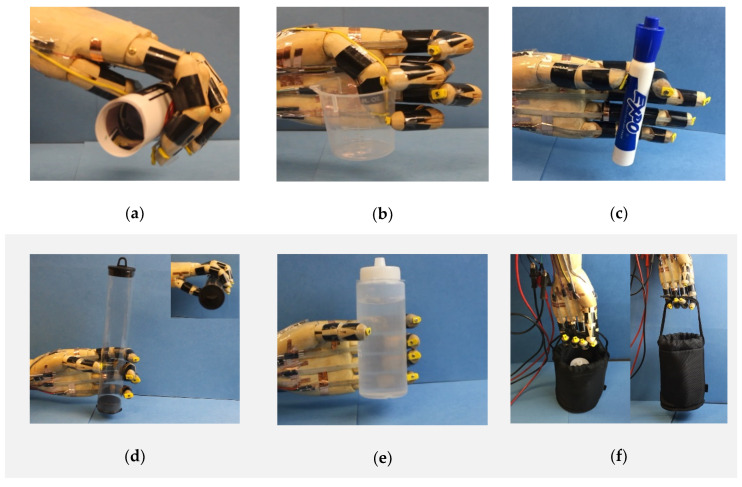
The biomimetic prosthetic hand can hold objects of different sizes and shapes and can carry different weights. (**a**) Holding a paper tube with a diameter of 30 mm. (**b**) Grabbing a plastic beaker. (**c**) Picking up a marker pen with a diameter of 20 mm. (**d**) Carrying a plastic tube with a weight of 2.12 g. (**e**) Holding a bottle of water with a weight of 200 g. (**f**) Lifting a basket with a weight of 252 g.

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
