# Peer review of "Biomimetic Prosthetic Hand Enabled by Liquid Crystal Elastomer Tendons"

_micromachines, 2021, doi:10.3390/mi12070736_

Round 1
Reviewer 1 Report
The authors presented a study using liquid crystal elastomer combined with liquid metal as artificial tendon for the design of a biomimetic prosthetic. The authors measured the response of the LCE tendon to electric current, contraction strain and contraction stress, and shown the capabilities of the design within some practical applications. Technically, the manuscript is very interesting and potentially show novelty, however, the current format does not reflect the novelty and technological breakthrough of the work. As such I consider it suitable for publication only after appropriately addressing the suggestions below.
- Precedent work in the field of LCEs is missing and should be considered to improve the significance such as works from Parmeggiani, Hegmann and Sanchez-Ferrer need to be discussed and added.
- The base mechanics that invokes the functionality of prosthetic hand is temperature which induces the contraction. In this study, the temperature of actuation appears higher than expected; for a prosthetic hand to function as such, the temperature range should be within physiological temperature. Could the authors provide comments on how this system and the referenced temperature range would apply to practical applications?
- The LCE preparation must be described in more details in the SI or be supported by a reference if the work is using a similar LCE preparation.
- Some figures require additional descriptions. i.e. the caption in Figure 2b and d need further clarification. Offering an accelerated or more accelerated video experience in SI would improve the conveyed concept.
Author Response
Point 1: Precedent work in the field of LCEs is missing and should be considered to improve the significance such as works from Parmeggiani, Hegmann and Sanchez-Ferrer need to be discussed and added.
Response 1: We thank the reviewer for the constructive comments, which have helped us to improve the overall quality of the manuscript. We have cited relevant literature and added discussion in introduction.
Modifications to the manuscript:
In line 57, we added the following statement marked in blue.
Because of their excellent mechanical characteristics, previous studies have shown interesting applications of LCEs, including actuation [81], cell culture [82] and tissue repair [83].
Point 2: The base mechanics that invokes the functionality of prosthetic hand is temperature which induces the contraction. In this study, the temperature of actuation appears higher than expected; for a prosthetic hand to function as such, the temperature range should be within physiological temperature. Could the authors provide comments on how this system and the referenced temperature range would apply to practical applications?
Response 2: We thank the reviewer for pointing this out. In practical applications, an insulation layer could be added between the LCE and the body. On the other hand, in rehabilitation applications, appropriate temperature increase can speed up blood circulation and help patients to recover.
Point 3: The LCE preparation must be described in more details in the SI or be supported by a reference if the work is using a similar LCE preparation.
Response 3: We have added references to the preparation of LCE.
Modifications to the manuscript:
At the end of the line 60, we added the following statement marked in blue.
The LCEs were prepared using procedures described in previous studies [68, 84-86].
Point 4: Some figures require additional descriptions. i.e. the caption in Figure 2b and d need further clarification. Offering an accelerated or more accelerated video experience in SI would improve the conveyed concept.
Response 4: We added clarifications to the figure captions: “(b) Temperature distribution and length change of a LCE tendon at the initial state (left, room temperature) and full contraction state (right, heated).” and “(d) The maximum contraction strain can be reached in LCE tendons at different electric currents.” We also added a video (the new video S1) to show the LCE contraction process with 2A current.

Reviewer 2 Report
This article reports on implementing liquid crystal elastomers in conjunction with liquid metal electrodes to design a tendon like actuators. These are incorporated into a prosthetic hand to drive finger movement via joule heating produced by passing electric current through the imprinted electrodes. Different hand gestures are showcased, together with a constant actuation of the tendon performed by altering the current to achieve constant temperature and not overheating the composite material.
While the overall idea is interesting and the material is wisely chosen and assembled into a working tendon, the manuscript is poorly written and greatly lacks proper characterization of the material and/or the prosthetic hand. The manuscript reads more as an internal work report rather than a research article. Hence, I do not recommend the publication of this manuscript.
I have below written a few comments and concerns regarding the presented work.
There is no accurate description on the materials and synthesis methods used. Authors mention ‘liquid metal’ and ‘very high bond tape’… From which material? Which manufacturer supplied the material and chemicals? Citations on the synthesis method is also missing. Authors mention they programmed the specimen by straining from L1 to L2. What are the usual strains used? Are they the same for each sample?
There is also insufficient information on the liquid crystal elastomer to be able to properly interpret the results. No thermomechanical/mechanical measurements or at least phase transition temperatures are provided that could put the experimental temperatures and the achieved mechanical stress into some context.
Air cooling is introduced at one point, but the air speed is provided for fan cooling only. I imagine that blown compressed air has different air speeds at different distances from the sample. Maybe a relaxation time vs. air speed graph would be more appropriate in figure 4.
Presented data also lacks information:
Figure 1 uses a photograph of a robotic hand and a schematic of a human hand. If the pictures do not originate from the author’s laboratory, a source or citation must be provided to give credit to the creators.
In figure 2c, was the heating turned off to enable the temperature to cool down after time?
In figure 2e, how was the maximum stress determined? I see no saturation of the stress (except for 1.5A), only a cut-off curve.
In figure 3, I would suggest to add a current vs. time diagram to understand more easily the experiment and what the numbers in the legend mean.
If the focus of this study is also the performance of the hand, I would suggest a more experimental take on the characterization of its capabilities, such as showcasing the speed of actuation, force exerted by the finger, etc. If it serves only as a demonstrator, then, as already mentioned, much more experimental focus should be provided on the LCE tendon.
Author Response
Point 1: There is no accurate description on the materials and synthesis methods used. Authors mention ‘liquid metal’ and ‘very high bond tape’… From which material? Which manufacturer supplied the material and chemicals? Citations on the synthesis method is also missing. Authors mention they programmed the specimen by straining from L1 to L2. What are the usual strains used? Are they the same for each sample?
Response 1: We thank the reviewer for the constructive comments, which have helped us to improve the overall quality of the manuscript. The liquid metal was made up of 25% gallium and 75% indium. The VHB was from 3M. The synthesis method was following previous studies, and references have been added (68, 84-86). The strain used for programming LCEs was fixed at 200%.
Modifications to the manuscript:
In lines 62 to 63, and 76 to 77 we added the following statement marked in blue.
It consists of a very high bond (VHB, 3M) tape support and bonding layer (10 mm wide and 25 µm thick) at the bottom, a liquid metal (LM) (25% gallium and 75% indium)…
In line 60-61 we added the following statement marked in blue.
The LCEs were prepared using procedures described in previous studies [68, 84-86].
At the end of line 92 we added the following statement marked in blue.
In the following, the LCE tendons were prepared with an original length of L1=40 mm, and were stretched by 200% to L2=120 mm during programming.
Point 2: Air cooling is introduced at one point, but the air speed is provided for fan cooling only. I imagine that blown compressed air has different air speeds at different distances from the sample. Maybe a relaxation time vs. air speed graph would be more appropriate in figure 4.
Response 2: The speed and temperature of compressed air are different from those of a fan. And we kept a constant distance between the samples and the compressed air outlet in order to minimize the effect of air speed difference. In this study, our purpose is to find the methods of cooling down the LCEs quickly. Through experiment comparison, we found that compressed air can cool down the LCEs most quickly among the three methods that can be easily accessed. We will conduct a more thorough study about the influence of air speed on the cooling of LCE in subsequent studies.
Point 3: Figure 1 uses a photograph of a robotic hand and a schematic of a human hand. If the pictures do not originate from the author’s laboratory, a source or citation must be provided to give credit to the creators.
Response 3: We have added the references in the figure captions.
Point 4: In figure 2c, was the heating turned off to enable the temperature to cool down after time?
Response 4: Yes, we cut off the current after reaching the maximum temperature.
Modifications to the manuscript:
At the end of the line115, we added the following statement marked in blue.
The current was then cut off after the LCEs reached the maximum temperatures.
Point 5: In figure 2e, how was the maximum stress determined? I see no saturation of the stress (except for 1.5A), only a cut-off curve.
Response 5: During the experiment, when the current was 2A and 2.5A, the maximum temperature reached 120℃ and 160℃ respectively (figure 2c). If the two ends of a LCE were fixed, the LCE would break when the temperature exceeded 120℃, so it did not reach the equilibrium state.
Point 6: In figure 3, I would suggest to add a current vs. time diagram to understand more easily the experiment and what the numbers in the legend mean.
Response 6: In Figure 3, we have added Figure 3c to show current vs time in three different control strategies.
Point 7: If the focus of this study is also the performance of the hand, I would suggest a more experimental take on the characterization of its capabilities, such as showcasing the speed of actuation, force exerted by the finger, etc. If it serves only as a demonstrator, then, as already mentioned, much more experimental focus should be provided on the LCE tendon.
Response 7: We thank the reviewer for the constructive comments. In this paper, we tried to demonstrate an effective method of actuating LCE tendon based prosthetic hand. In this system, there are many interesting problems need to be investigated, including characterization and optimization of the tendon actuation, and design and optimization of the hand performance. Due to the limit on length and scope of this paper, we are not able to address all the issues. However, we do plan to conduct more thorough investigations on this system in the future.

Round 2
Reviewer 2 Report
Many of the suggested improvements and corrections have been made to the article. All the information is now provided and more clearly presented, thus improving the overall readability. I agree with the publication.